

# A sentiment analysis approach for travel-related Chinese online review content

Hanyun Li, Wenzao Li, Jiacheng Zhao, Peizhen Yu and Yao Huang

Chengdu University of Information Technology, Chengdu, China

## ABSTRACT

Using technology for sentiment analysis in the travel industry can extract valuable insights from customer reviews. It can assist businesses in gaining a deeper understanding of their consumers' emotional tendencies and enhance their services' caliber. However, travel-related online reviews are rife with colloquialisms, sparse feature dimensions, metaphors, and sarcasm. As a result, traditional semantic representations of word vectors are inaccurate, and single neural network models do not take into account multiple associative features. To address the above issues, we introduce a dual-channel algorithm that integrates convolutional neural networks (CNN) and bi-directional long and short-term memory (BiLSTM) with an attention mechanism (DC-CBLA). First, the model utilizes the pre-trained BERT, a transformer-based model, to extract a dynamic vector representation for each word that corresponds to the current contextual representation. This process enhances the accuracy of the vector semantic representation. Then, BiLSTM is used to capture the global contextual sequence features of the travel text, while CNN is used to capture the richer local semantic information. A hybrid feature network combining CNN and BiLSTM can improve the model's representation ability. Additionally, the BiLSTM output is feature-weighted using the attention mechanism to enhance the learning of its fundamental features and lessen the influence of noise features on the outcomes. Finally, the Softmax function is used to classify the dual-channel fused features. We conducted an experimental evaluation of two data sets: tourist attractions and tourist hotels. The accuracy of the DC-CBLA model is 95.23% and 89.46%, and that of the F1-score is 97.05% and 93.86%, respectively. The experimental results demonstrate that our proposed DC-CBLA model outperforms other baseline models.

# INTRODUCTION

With the rise of tourism-based Internet products, tourism and Internet information are closely intertwined, and the Internet has become one of the most critical tools for analyzing tourism activities. More and more travelers are sharing images and videos of their travel experiences on developing social platforms such as microblogs, Twitter, and travel websites. Internet users can comment and share, resulting in many online travel review texts with significant potential value (*Alamanda et al., 2019*). Most travelers rely on online evaluations when planning their trip itineraries. They use these evaluations to assess the advantages and

Corresponding author
Wenzao Li, lwz@cuit.edu.cn

disadvantages of attractions and hotels, aiming to choose a more affordable, convenient, and enjoyable travel plan. In addition, online reviews have a higher impact on vacation planning than personal recommendations. By analyzing and processing visitors' opinions, the tourism industry can gain valuable insights into visitors' emotional preferences. These insights enable the tourism industry to make improvements and promptly identify issues, enhancing visitor satisfaction. Therefore, accurately mining passengers' opinions benefits the development of the tourism business as a whole (*Alaei, Becken & Stantic, 2019*).

The majority of early research utilized sentiment analysis techniques relying on sentiment lexicon and machine learning. With the advancements in deep learning technology, neural network-based approaches have become widely adopted. Current research is concentrating on utilizing deep learning models to process tourism texts, aiming to extract more comprehensive semantic information. *Yan et al. (2019)* proposed a method for filtering the seed set of a sentiment lexicon based on the word vector model. In this method, sentiment words are represented as vectors. The distance between these vectors is measured, and this measurement is used to establish the selection criteria and classification foundation for the seed word set. Finally, a sentiment lexicon comprising online evaluations of mountain tourism destinations was compiled. This approach to the tourism sentiment lexicon is simple and widely applicable. However, it heavily relies on the quality of the lexicon. Creating a comprehensive tourism lexicon for abundant tourism texts is challenging due to the significant amount of labor required. *Wan, Jiazhen & Zhang (2021)* proposed a text sentiment analysis method that integrates an improved stacking algorithm with rules. The method enhances the stacking algorithm and combines it with the text rule method. Experiments were carried out on three distinct groups of Web comment texts. The results of these experiments demonstrate the method's strong performance and high accuracy in sentiment analysis of Web comment text. However, it still requires the contrived selection and extraction of features, ignores the correlation between features, and has limited generalizability and scalability.

Deep learning is an emergent subfield of machine learning algorithms. The vast majority of studies construct sentiment classifiers using deep neural networks, such as convolutional neural networks (CNN), recurrent neural networks (RNN), and attention mechanism networks, among others. Deep learning-based methods can examine the relationship between target words and sentiment polarity terms in sentences without the need for manually compiling sentiment lexicon (*Li et al., 2020*). In addition, it can overcome the limitations of conventional machine learning algorithms by automatically learning the semantic aspects of the text, in contrast to systems that require human intervention in feature selection. *Cai et al. (2021)* proposed a Bidirectional Encoder Representations from Transformers (BERT)-based end-to-end opinion mining approach for tourist reviews. First, BERT is used to encode travel reviews. Then the corresponding travel reviews are sequentially annotated after decoding by the downstream pointer network to obtain an opinion word and category binary to form a complete opinion representation. These deep learning techniques make affective computing more applicable and improve its generalizability. *Vernikou, Lyras & Kanavos (2022)* proposed a BERT-bidirectional long short-term memory (BERT-BiLSTM) model for classifying user sentiment based on

COVID-19-related Twitter posts. The BiLSTM module captures global sequence features of text, but its loop mechanism results in inefficient training. *Huang, Lin & Wang (2022)* proposed a BiLSTM network based on the attention mechanism (BiLSTM-ATT) model. The BiLSTM is used to capture contextual and semantic information. In addition, the attention module boosts the impact of essential sentiment words on the recognition results and mitigates the interference of irrelevant features. *Guo et al. (2021)* proposed a sentiment classification model of CNN with a semantic initialization filter to address the sentiment classification problem of microblog comments. The CNN module extracts the local semantics of the text. However, Chinese expressions are diverse, and it is not uncommon to encounter oversized metaphors, sarcasm, and excessively colloquial phrases in travel reviews. These discourses of varying quality make identifying affective tendencies challenging. CNN-based methods utilise convolutional kernels of uniform size to extract features, which are effective in extracting local features within the text. However, this approach may result in a loss of feature position or order in the pooling layer, thereby posing challenges in capturing semantic features and long-distance dependencies. RNN-based methods pass information through a cyclic structure, making them adept at extracting contextual information. However, they face challenges in deriving local features from the text (*Birjali, Kasri & Beni-Hssane, 2021*). To improve the learning ability of the proposed model and the precision of emotion recognition, we propose a dual-channel model that combines a dual-channel CNN and BiLSTM-ATT, named DC-CBLA. It not only exploits the advantages of CNN and long short-term memory (LSTM) to extract text features but also introduces a multi-channel attention mechanism on top of that. By introducing the attention mechanism, different weights can be assigned based on the impact of various words on the classification results, thereby improving the performance of the text classification model. The main contributions of this article are summarized as follows:

1. This article uses the BERT pre-training model to improve the multi-word sense problem encountered in Word2Vec. BERT captures rich contextual information using a bidirectional transformer encoder with an attention mechanism, resulting in more accurate and context-aware word vectors. Based on the experimental results, the BERT word vector model outperforms the traditional word embedding model in terms of performance metrics.

2. To enhance the model's understanding of the text's semantics, we extract local and global features from the obtained word vectors using CNN and BiLSTM. Additionally, we employ the attention mechanism to assign greater weight to features with higher impact, thus emphasizing crucial information in the text and reducing the interference of noisy features.

3. The output information from the dual-channel is combined to create the final vector representation of text information, and the text classification is then performed based on this vector.

4. We conducted experiments on the tourist attraction and hotel datasets separately. In addition, we explore the effect of word pre-training models on classification results and the tuning of hyperparameters.

## RELATED WORK

There are three main approaches to sentiment analysis: the approach of sentiment lexicon, the approach of traditional machine learning, and the approach of deep learning (*Alantari et al., 2022*).

### The approach of sentiment lexicon

In the early stages of sentiment analysis research, experts in related disciplines and sentiment analysis researchers developed a comprehensive sentiment lexicon. They also established reasonable rules for calculating sentiment values. These rules were used to determine the polarity of sentiment. *Rao et al. (2014)* employed three pruning strategies to construct a lexicon for socio-emotional detection automatically. Additionally, they proposed a topic-based modeling approach to build a topic-level lexicon, where each topic is associated with social emotions. *Han et al. (2018)* proposed a new domain-specific lexicon generation method. It uses the unlabeled comment dataset to generate domain-specific sentiment lexicons based on a SentiWordNet (SWN)-based sentiment classifier. This approach relies excessively on the establishment of sentiment lexicons, which are rarely updated to meet the rapid generation of new words in the information age. Moreover, the sentiment lexicon-based approach consumes a large amount of human and material resources.

### The approach of traditional machine learning

Due to the enormous amount of tourism review texts, it is labor-intensive and time-consuming to analyze them only relying on traditional manual methods. Moreover, this approach significantly reduces the accuracy of recognition. More and more academics are applying machine learning algorithms to travel sentiment analysis as machine learning theory continues to advance. The maximum entropy model, the Naive Bayesian model (NB), and the support vector machine (SVM) are frequently employed for sentiment analysis tasks. *Liang, Liu & Zhang (2020)* introduced the Document-to-vector (Doc2vec) approach combined with SVM to more effectively extract valid information from text for text sentiment classification. To minimize noise interference, they normalized the dataset by removing irrelevant text information. Then, Doc2Vec was used to perform word quantization, enabling the acquisition of a vector space with a high degree of diversity in semantic expressions. Finally, microblog comments are analyzed using the SVM classifier, which is preferable to the sentiment lexicon. *Elgeldawi et al. (2021)* utilized five methods: Grid Search, Random Search, Bayesian Optimization, Particle Swarm Optimization, and Genetic Algorithms, to tune the hyperparameters of machine learning models and conducted a comprehensive comparative analysis. The goal was to address the limitations of existing literature, which often relies solely on Grid Search and Random Search for parameter tuning. *Al-Hadhrami, Al-Fassam & Benhidour (2019)* proposed a multiple-sentiment lexicon to reduce the dimensionality of feature and label matching. The Term Frequency-Inverse Document Frequency (TF-IDF) feature weighting method was used to calculate the weights. According to experimental findings, uni-grams and linear SVM models are appropriate for use in English Twitter sentiment analysis.

## The approach of deep learning

Techniques for sentiment analysis based on machine learning depend excessively on the high quality of the corpus. The contextual order cannot be distinguished effectively, and their classification performance is marginally impaired. Deep learning techniques improve this problem by automatically learning text sentiment from various samples and automatically conducting feature representation, which provides new ideas for travel data mining and classification. With the assistance of the Word2vec model, high-dimensional word sequences are transformed into low-dimensional word vectors, which contain substantial contextual semantic information (*Mikolov et al., 2013*). This makes neural network models a popular technique for classifying individuals' emotions. *Kim (2014)* transformed preprocessed word vectors into two-dimensional word vector matrices and used convolutional kernels of different sizes to get local features from comments. The pooling layer extracts the most apparent features to achieve sentence-level text classification. However, convolutional neural networks do not effectively handle serialized information. Meanwhile, the recurrent neural network proposed by *Zaremba, Sutskever & Vinyals (2014)* has been used in natural language processing tasks such as text classification because its structure is excellent at processing serialized information. LSTM is an improved RNN that reduces the occurrence of gradient disappearance and explosion and can better extract semantic information. *Wang (2022)* divided tourism review texts into various topics. Then they proposed an improved LSTM framework for rural tourism topic feature extraction, which has substantial improvements in each index over the traditional RNN algorithm. *Li & Ning (2020)* proposed a CNN-LSTM model for the categorization of Chinese text. The text was initially converted to a low-dimensional using word embedding and then extracted using CNN and BiLSTM. The model achieved an accuracy of 98.03%. *Wang & He (2022)* proposed a neural network classification model incorporating emojis. The model extracts deep context from bi-directional gated recurrent unit (BiGRU). The emoji and text vectors are then put into an attention mechanism. It assigns greater weight to the vector information that affects the classification result. *Zeng, Yang & Zhou (2022)* proposed a hybrid model of latent Dirichlet allocation (LDA) and BiLSTM-ATT to analyze the evolution of public opinion regarding prevalent topics on Weibo. Python's Scrapy framework is utilized to crawl textual comments about the events, and BiLSTM is employed for text sentiment analysis. The results show that the hybrid model constructed based on LDA and BiLSTM effectively captures the trending topics and temporal changes in sentiment during the events. It also reveals the main reasons for the outbreak of the events, the primary topics during the event dissemination phase, and the outcomes of the event processing.

Table 1 describes the various sentiment analysis methods used in the past. Traditional sentiment analysis methods lack deep semantic support for sentiment judgments in reviews. Moreover, due to the sparse features and thin textual and contextual relationships characteristic of travel texts, the existing sentiment classification methods have disadvantages, such as low accuracy and slow model convergence. Insights from constructing sentiment analysis models based on different word embedding vector approaches and neural network models. Our model employs a BERT pre-training model to extract dynamic word vector representations that correspond to the contextual

**Table 1  Summary of sentiment analysis methods from past research work.**

| References | Methodology | Dataset | Number of data |
|---|---|---|---|
| *Rao et al. (2014)* | Sentiment pruning strategies to improve word-level sentiment lexicons. | Sina Dataset, SemEval-2007 Dataset | 1,854,718, 109,129 |
| *Han et al. (2018)* | The unlabeled reviews are used to generate domain-specific sentiment lexicon | Movie Review Dataset, Amazon Dataset | 50,000, 1,140,496 |
| *Liang, Liu & Zhang (2020)* | After regularizing the dataset and using the Doc2vec+SVM method | Chinese Opinion Analysis Evaluation (COAE) 2013, COAE2015 | 22,890, 30,049 |
| *Elgeldawi et al. (2021)* | Five hyperparameter methods for hyperparameter tuning of machine learning algorithms | Hotel Review Dataset | 7,000 |
| *Al-Hadhrami, Al-Fassam & Benhidour (2019)* | Sentiment weights are calculated using TF-IDF and classified using machine learning. | Sentiment140 Dataset | 1,578,612 |
| *Kim (2014)* | CNN for extracting word character features | Movie Review (MR) Dataset, Stanford Sentiment Treebank (SST)-1 Dataset, SST-2 Dataset, Subjectivity Dataset, TREC Question Dataset, Customer reviews Dataset, Multi-Perspective Question Answering (MPQA) Dataset | 10,662, 11,855, 9,613, 10,000, 5,952, 3,773, 10,604 |
| *Zaremba, Sutskever & Vinyals (2014)* | LSTM extracts semantic information | Penn Tree Bank (PTB) Dataset | 900,000 |
| *Wang (2022)* | Improved LSTM framework for tourism theme feature extraction | Rural Tourism Dataset | 18,566 |
| *Li & Ning (2020)* | CNN and LSTM hybrid models are used to derive both local and global features | THUCNews Dataset | 740,000 |
| *Wang & He (2022)* | BiGRU is used to extract global information, while the attention mechanism performs feature weighting to emphasize important features | Weibo Sentiment 100k Binary Dataset | 119,988 |
| *Zeng, Yang & Zhou (2022)* | The topics are classified by LDA, and the text features are extracted by BiLSTM-ATT | COVID-19 Review Dataset | 30,525 |

representation, thereby improving the precision of the vector semantic representation. We then fuse the advantages of CNN and BiLSTM in sentiment analysis tasks to extract text features. In addition, an attention mechanism is used to extract more essential semantic features. The experimental results demonstrate that our approach can effectively enhance the model's overall structure and functionality.

## METHODOLOGY

This section describes our proposed DC-CBLA framework architecture and the key algorithms that make up the model.

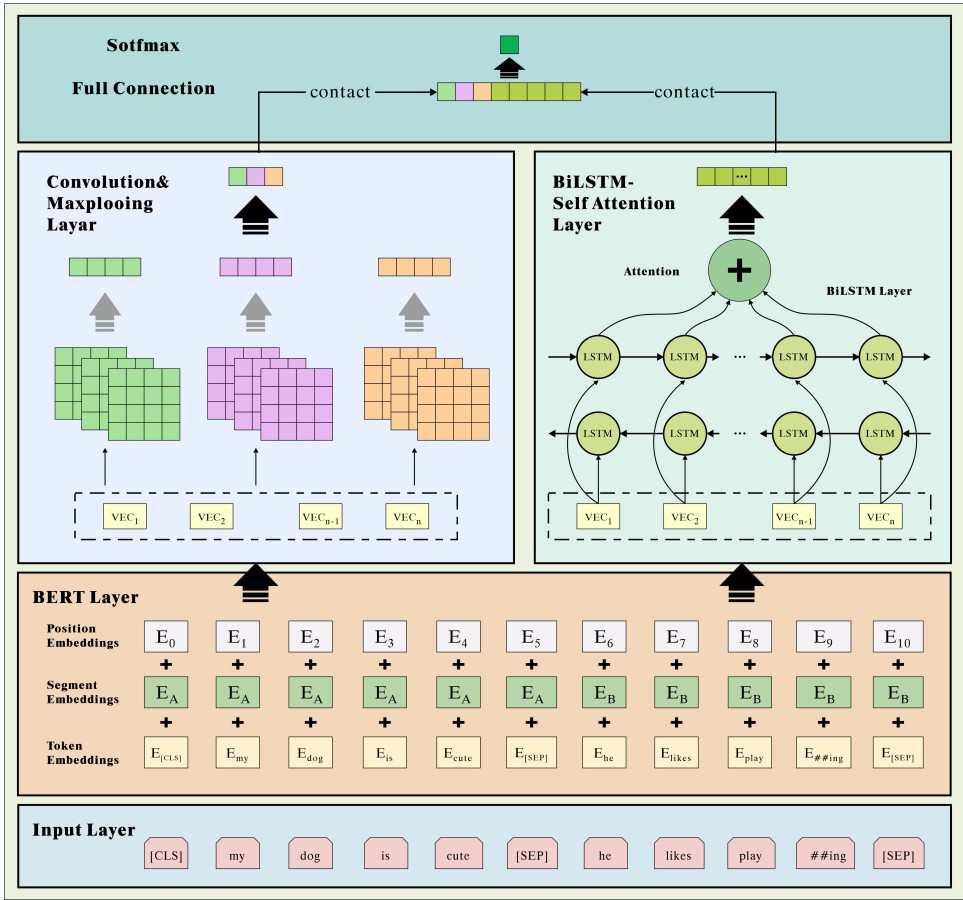

**Figure 1  Model overall architecture diagram.**

## Model architecture

CNN-based methods typically utilize a sliding window convolution operation to extract local features of the text. In sentiment analysis, some sentiment terms may be scattered throughout the text rather than being restricted to a particular location. Consequently, CNNs lack the ability to specifically focus on sentiment terms within the text. When there are semantic dependencies exist in the text, classification accuracy decreases, but the CNN network can better extract the text's local features and semantic information (*Ramaswamy & Chinnappan, 2022*). The BiLSTM-ATT model integrates information from preceding and subsequent texts, which is beneficial for the extraction of text's deep-level features. Combining the complementary advantages of both, we propose a dual-channel hybrid divine network model for comprehensive and detailed text relevance extraction. As depicted in Fig. 1, the DC-CBLA model comprises three main parts: the BERT layer, the feature extraction layer, and the fully connected layer.

To better extract the underlying text's lexical, semantic, and contextual feature information. We use the BERT method for word vectorization. First, the text of the travel review is transmitted through an internal word embedding layer for static word vector

encoding on a word-by-word basis. Then it is sent to the BERT layer, and the dynamic word vector representation corresponding to the context is generated in conjunction with the context. After obtaining the word vector matrix, we employ dual-channel CNN and BiLSTM with a fused attention mechanism in the hybrid neural network layers to extract local and global features of the travel text, respectively. The CNN network consists of three convolutional layers with the same number of convolutional kernels but varying sizes. The pooling layer for feature descending and splicing can extract more local semantic features. BiLSTM models the spatial semantic information based on forward LSTM and backward LSTM networks, which can better capture the context's semantic information and derive the global characteristics of the travel review text. In addition, the attention mechanism is employed to compute weight assignment coefficients, enabling a higher emphasis on crucial components of the text and enhancing feature representation. In the fully connected layer, the proposed local and global features are fused, and the fused features undergo dimension reduction. The prediction vector is then input into the Softmax classifier to perform the classification.

## BERT model in sentiment analysis

There are numerous multiple-word meaning issues in travel reviews; therefore, it would be inappropriate to use a globally uniform representation of word vectors, and the addition of external knowledge would make the model somewhat less efficient to train. In order to generate dynamic word vectors with contextual meanings, we employ the BERT pre-training model. The dual transformer encoder is the most indispensable component of BERT. Figure 2 depicts the construction. Typically, the key for a traditional encoder–decoder to figure out the sequence question is realizing the construction in CNN and RNN. CNN's convolution operation is unsuitable for sequentially parsing text, and RNN's structure is not parallelizable and runs slowly. The encoder of the transformer employs a multi-headed attention mechanism and feedforward neural network, which effectively resolves the long-term dependency problem and enhances the model's recognition capability (*Devlin et al., 2018*). The calculation formula is:

$$Attention(Q, K, V) = Softmax(\frac{QK^T}{\sqrt{d_k}})V \tag{1}$$

where $Q$, $K$, and $V$ are the matrices formed by the linear mapping of the input vectors, which are obtained by the matrix product of a random initialization matrix with the vectors of the embedding layer, indicating the dimensionality of the vectors. $d_k$ is the dimension of the input vector.

## LSTM and BiLSTM in sentiment analysis

*Hochreiter & Schmidhuber (1997)* proposed the LSTM to solve the problem of gradient explosion due to the long-term dependence of RNNs generated by processing too much information. The cell was added to the LSTM, and the memory function was realised by controlling the transmission state through the gating state, effectively solving the gradient explosion problem. The model structure is illustrated in Fig. 3. The LSTM network cell
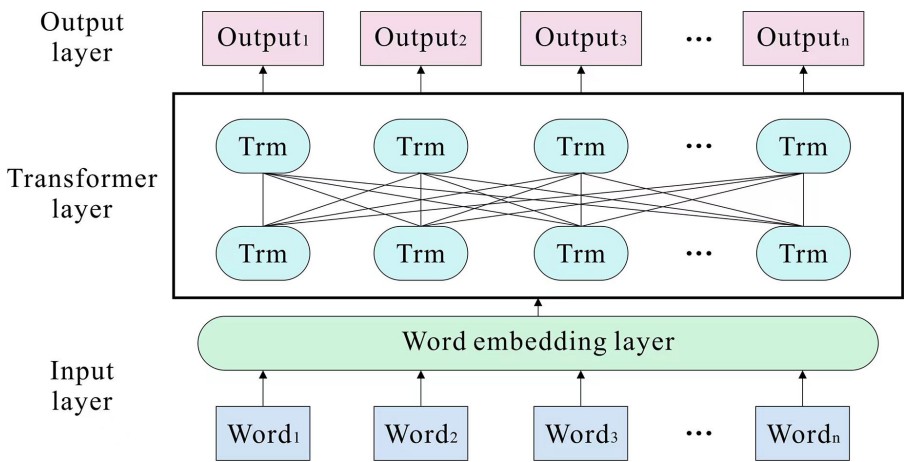

Output layer

Transformer layer

Input layer

**Figure 2  BERT model structure.**

consists of an input gate $i$, a forget gate $f$, and an output gate $o$. Equations (2) through (6) illustrate the specific computation process of LSTM units.

$$i_t = \sigma(W_i \cdot [h_{t-1}, X_t] + b_i) \tag{2}$$

$$o_t = \sigma(W_o \cdot [h_{t-1}, X_t] + b_o) \tag{3}$$

$$f_t = \sigma(W_f \cdot [h_{t-1}, X_t] + b_f) \tag{4}$$

$$c_t = f_t * c_{t-1} + i_t * tanh(W_c) \cdot [h_{t-1}, X_t] + b_c \tag{5}$$

$$h_t = o_t * tanh(c_t) \tag{6}$$

where $t$ represents time, $x_t$ represents the input at time $t$, $h_t$ represents the hidden layer output information, and $h_{t-1}$ represents the hidden layer output information of the previous moment. $*$ denotes element multiplication. $W_i$ and $b_i$ are the input gate's parameters; $W_f$ and $b_f$ are the forgetting gate's parameters; $W_o$ and $b_o$ are the output gate's parameters. $c_{t-1}$ and $c_t$ represent the previous cell's state and the current cell's state, respectively.

Texts describing travel reviews contain extensive text sequences. However, because LSTM networks propagate in a single direction, words expressing their sentiments can appear anywhere in the text during a text sentiment classification task. This leads to the problem that the later words in the sequence are more significant than the earlier ones. The BiLSTM neural network consists of forward-running LSTM units and backward-running LSTM units. These units capture information from both directions of context and effectively learn the long-term dependencies of text. Therefore, we use BiLSTM to extract global features.

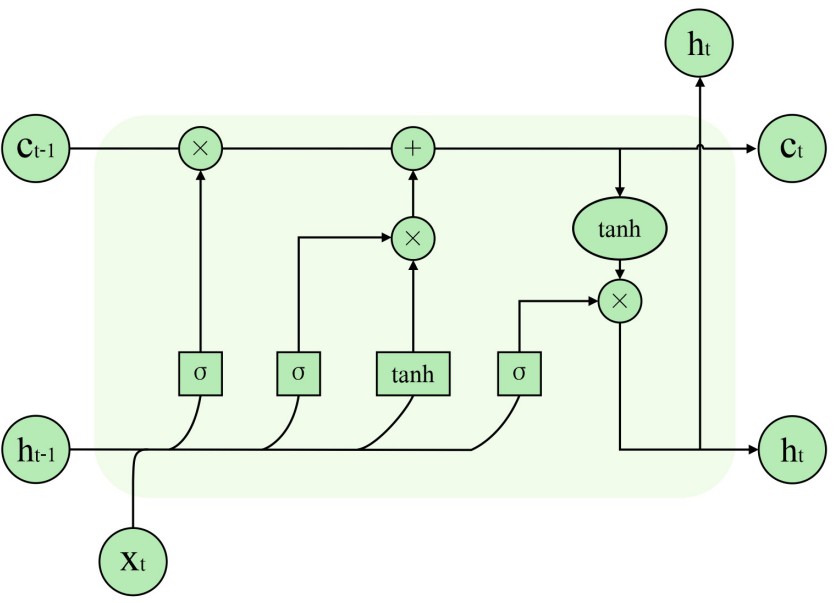

**Figure 3 LSTM model structure.**

## CNN model in sentiment analysis

CNN is fast to train and has an excellent performance in the field of short texts (*Feng & Cheng, 2021*), and travel texts contain an enormous amount of short texts; therefore, CNN is used to extract local information from the text. Figure 4 depicts the architecture. CNN consists of three parts: convolutional layer, pooling layer, and fully connected layer. After the word embedding layer, the input sequence is $S = \{x_1, x_2, x_3, \ldots, x_n\}, x \subset R^{h \times d}$ where $n$ is the length of the sentence and $d$ is the vector text. The convolutional layer is given a convolutional kernel of size: $w \in R^{h \times d}$ and a filter window in the sequence $w_{i:i+h-1}$ matrix to perform the convolution operation to extract the local features. The process of extracting each local feature $c_i$ as follows:

$$c_i = f(w \times x_{i:i+h-1} + b) \tag{7}$$

where $f$ represents the nonlinear activation function ReLU, $b$ is a bias term, and $x_{i:i+h-1}$ represents word vectors in different positions. After the convolution operation, the feature sequence is obtained $S = \{c_1, c_2, c_3, \ldots, c_{n-h+1}\}$. The maximum pooling layer retains the most extensive features, and the input objects are reduced in dimensionality to reduce the number of feature vectors and network parameters to prevent overfitting. Finally, the resulting features are stitched together and fused into a vector by a fully connected layer.

## Attention mechanism in sentiment analysis

We introduce an attention mechanism for learning the relative importance of various words to increase the influence of crucial sentiment words on the overall sentiment expression. The use of an attention machine enhances the performance of the model by allowing

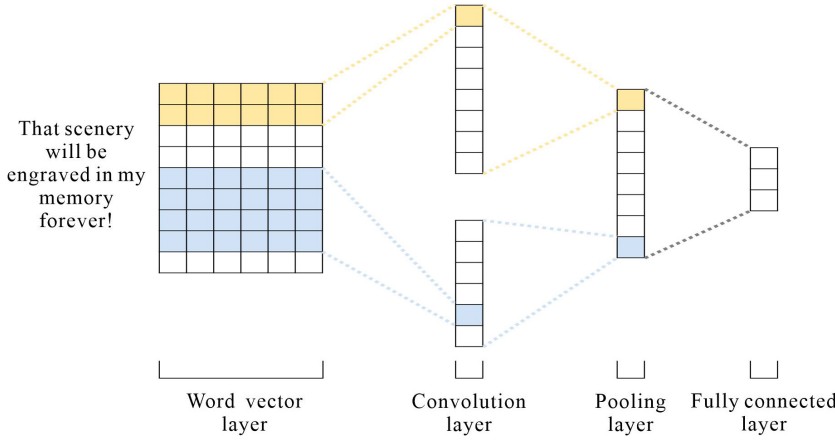

**Figure 4  CNN model structure.**

for more critical information (*Abudouwaili et al., 2022*). In this experiment, weights for emphasizing keywords are assigned using a feed-forward neural attention strategy. First, calculate the attention weight score $e_i$:

$$e_i = tanh(W_a h_t + b_0) \tag{8}$$

where $W_a$ represents the weight matrix, $b_0$ represents the bias coefficient. $h_t$ is the output of the BiLSTM.

$$u_t = \frac{exp(e_i)}{exp\sum_{t=1}^{m} e_i}. \tag{9}$$

Finally, the output vector $h_t$ of the BiLSTM layer and the weight vector $u_t$ are subjected to dot product and accumulation operations to obtain the output of the $a_t$. This approach can assign corresponding attention weights to the output of the hidden layer according to the weight size, which constitutes a weighted semantic vector representation of the feature vector. It can enhance the feature representation of text sequences that describe tourist hotels and attractions.

$$a_t = \sum_{t=1}^{n} u_t h_t. \tag{10}$$

## EXPERIMENTAL PROCESS AND RESEARCH

### Experimental data

Our experimental data are categorized into two groups for this experiment: tourist attractions and tourist hotels. Data on tourist attractions were collected from all 5A-rated tourist attractions in Sichuan, including the Dujiangyan Dam, Wuhou Temple, and Jinsha Ruins Museum. Hotel information is obtained from popular travel websites. The quality of the raw data collected from online crawling is inconsistent. Therefore, we removed invalid data, such as overly brief messages and invalid comments. Additionally, we eliminated

**Table 2 Statistical information on data sets.**

| Designation | Total | Positive | Negative |
|---|---|---|---|
| Tourist attractions | 35,804 | 26,748 | 9,056 |
| Tourist hotels | 58,372 | 49,501 | 8,871 |

**Table 3 Model parameters.**

| Hyperparameter | Value |
|---|---|
| CNN number of convolution kernels | 100 |
| CNN filter size | (3,4,5) |
| Leaning rate | 0.00005 |
| Dropout | 0.2 |
| Epoch | 8 |
| Batch size | 24 |
| BiLSTM hidden size | 256 |
| BiLSTM hidden layer | 2 |

special characters and emojis from the text. As a result, we obtained 35,804 reviews of tourist attractions and 58,371 datasets for travel hotels. The details of the dataset are shown in Table 2.

## Experimental parameter

In the training of deep learning models, different hyperparameter settings will have a great impact on the classification of the model. Hence, in this experiment, appropriate hyperparameters were selected based on a review of relevant literature. The window size of the convolution kernel is (3,4,5), and the number of convolution kernels is 100. After the convolution pooling operation, the output of each layer is fused to obtain a richer local feature below. The hidden layer of the BiLSTM layer has 256 neurons (*Luo & Zhang, 2022*). Due to the possible overfitting of our model with too numerous parameters, the dropout is set to 0.2 in order to prevent the phenomenon of overfitting lines in the CNN and LSTM network layers (*Ghosh et al., 2022*). Considering that a larger batch size would lead to insufficient memory consumption, while a smaller batch size would result in excessive training time, this experiment selects suitable batch size and epoch values based on the capacity of our experimental equipment. We set the batch size to 24 and conducted eight training rounds. The model hyperparameters are configured as presented in Table 3.

## Evaluation metrics

In this experiment, accuracy, precision, and F1-score were used to evaluate the predictive performance of the model. The calculation formula is as follows:

$$accuracy = \frac{TP + TN}{TP + TN + FP + FN} \tag{11}$$

$$precision = \frac{TP}{TP + FP} \tag{12}$$

$$recall = \frac{TP}{TP + FN} \tag{13}$$

$$F1 - score = \frac{2 * precision * recall}{precision + recall} \tag{14}$$

where true positives (TP) represent the number of predicted positive samples in positive sentiment samples, false positives (FP) represent the number of predicted positive samples in negative sentiment samples, false negatives (FN) represent the number of predicted negative samples in positive sentiment samples, and true negatives (TN) represent the number of predicted negative samples in negative sentiment samples.

## Experimental results and analysis

The experiments presented in this article establish multiple categories of comparison experiments, including the single network layer and the hybrid network. Through these comparison experiments, the effectiveness of the DC-CBLA model proposed in this article can be evaluated. The description of the baseline model for this experiment is as follows:

1. BERT (*Devlin et al., 2018*): It employs the final hidden vector generated by BERT to predict the sentiment of input sentences.

2. BERT-CNN (*Chen, Cong & Lv, 2022*): Based on the BERT model, CNN is added to further extract key feature information.

3. BERT-LSTM: Based on the BERT model, directly backward semantic modeling of the input sequence extracts features of the travel review text. Moreover, through the fully connected layer for dimensionality reduction.

4. BERT-BiLSTM (*Vernikou, Lyras & Kanavos, 2022*): Based on the BERT, add bi-directional LSTM and capture the features of front and back dependencies.

5. BERT-BiLSTM-ATT (*Guo et al., 2022*): Based on the BERT model, BiLSTM is utilized to capture pre- and post-text semantic information from the input sequence, enabling the extraction of high-level features from tourism text. The attention mechanism is then applied to perform feature weighting on text features, reducing the impact of noisy features on the classification performance.

6. BERT-BiGRU-ATT (*Ma et al., 2022*): Based on the BERT-BiLSTM-TT model, the BiLSTM structure is replaced by the BiGRU structure to solve the gradient message problem when the sequence is too long.

7. BERT-CNN-LSTM (*Khan et al., 2022*): Based on the BERT model, CNN first extracts the local features of the input sequence. The BiLSTM then extracts the forward and backward semantic information of CNN output to further construct the feature representation of tourism text.

8. DC-CBLA: This article's model.

A comparison of the experimental results between the BERT-LSTM model and the BERT-BiLSTM model demonstrates that BERT-BiLSTM outperforms BERT-LSTM in all evaluation metrics. As the unidirectional LSTM network fails to adequately capture the

semantic information of preceding and subsequent texts in the processing of travel review text sequences. The BiLSTM network consists of two layers of networks, forward LSTM and reversed LSTM, which can better capture the contextual information between the semantics of the utterances. Introducing the attention mechanism can make it easier for the BiLSTM network to extract the global features of the context. It reduces the effect of noisy features on the classification effect by weighting the features to improve the semantic representation. From Table 4, it can be observed that BERT-BiLSTM achieves a 0.49 % improvement in accuracy after integrating the attention mechanism. Gated recurrent unit (GRU) represents a new generation of recurrent neural networks that closely resembles LSTM. The accuracy of BERT-BiLSTM-ATT is improved by 0.3 %, and the F1-score is improved by 0.18 % compared with BERT-BiGRU-ATT, indicating that the BiLSTM network has better performance in the task of sentiment analysis of comments. This dataset contains a significant number of short texts. In addition, the contextual semantic relevance of travel short text data is not high, and the features are often redundant and not strongly correlated. CNN can obtain relatively excellent comprehensive model performance in the field of processing short texts. Therefore, BERT-CNN achieved excellent results in this experiment.

The DC-CBLA exhibits a significant improvement in precision, recall, and F1-score compared to the BERT-CNN and BERT-LSTM alone. As the DC-CBLA model effectively integrates their advantages, it can extract both local information and sentence-level contextual semantic information from travel reviews, leading to the generation of more comprehensive and richer travel text features. The BERT-CNN-LSTM can effectively extract contextual information. However, it assumes that each word has an equal impact on the final result and does not prioritize more important words during the classification process. In contrast, the DC-CBLA model not only effectively extracts features but also focuses on words that have a substantial impact on the classification outcome. Additionally, BERT-CNN-LSTM employs a recursive structure, which is more susceptible to gradient disappearance and explosion problems. In contrast, the DC-CBLA model extracts features using a parallel structure, effectively mitigating the gradient-related issues. Table 4 indicates that compared to BERT-CNN-LSTM, our proposed method improves accuracy by 0.32% and recall by 0.11%.

## Generalization ability analysis

If experiments are conducted on a single dataset only, the generalization capability is not guaranteed. In this experiment, we not only utilized tourist attraction data but also included tourist hotel data to validate the effectiveness of our model. The results of the tourist hotel experiment are presented in Table 5.

The accuracy variation of each model under eight epochs is plotted in Tables 6 and 7, respectively. The table shows that accuracy varies by epoch, with some models exhibiting early overfitting. The DC-CBLA model demonstrates fewer fluctuations in accuracy, a smoother relative trend, and a higher accuracy rate compared to the other baseline models. Furthermore, it outperforms the other baseline models in terms of accuracy during subsequent training, thus further validating the model presented in this article.

**Table 4  Tourist attraction data experiment results.**

| Model | Accuracy (%) | Precision (%) | F1-score (%) |
|---|---|---|---|
| BERT | 92.11 | 94.52 | 96.27 |
| BERT-CNN | 94.95 | 95.20 | 96.90 |
| BERT-LSTM | 92.41 | 94.60 | 96.39 |
| BERT-BiLSTM | 94.25 | 95.95 | 96.43 |
| BERT-BiLSTM-ATT | 94.74 | 96.15 | 96.73 |
| BERT-BiGRU-ATT | 94.44 | 96.33 | 96.55 |
| BERT-CNN-LSTM | 94.91 | 96.42 | 96.96 |
| DC-CBLA | 95.23 | 96.53 | 97.05 |

**Table 5  Tourist hotel data experiment results.**

| Model | Accuracy (%) | Precision (%) | F1-score (%) |
|---|---|---|---|
| BERT | 84.39 | 84.32 | 91.41 |
| BERT-CNN | 87.14 | 90.10 | 92.46 |
| BERT-LSTM | 84.65 | 84.79 | 91.47 |
| BERT-BiLSTM | 85.79 | 85.96 | 92.04 |
| BERT-BiLSTM-ATT | 86.78 | 87.66 | 92.68 |
| BERT-BiGRU-ATT | 86.58 | 87.55 | 92.37 |
| BERT-CNN-LSTM | 87.64 | 90.22 | 92.59 |
| DC-CBLA | 89.46 | 90.91 | 93.86 |

**Table 6  Variation in accuracy values for tests at different epochs on the travel hotel dataset.**

| Model | Epoch | | | | | | | |
|---|---|---|---|---|---|---|---|---|
| | 1 (%) | 2 (%) | 3 (%) | 4 (%) | 5 (%) | 6 (%) | 7 (%) | 8 (%) |
| BERT | 20.47 | 84.39 | 84.24 | 82.20 | 82.20 | 82.20 | 82.20 | 82.20 |
| BERT-CNN | 17.21 | 86.50 | 87.14 | 80.01 | 81.90 | 86.62 | 85.92 | 86.59 |
| BERT-LSTM | 27.25 | 84.65 | 83.32 | 83.20 | 83.20 | 83.20 | 83.20 | 83.20 |
| BERT-BiLSTM | 64.30 | 84.92 | 85.79 | 83.30 | 83.30 | 83.30 | 83.30 | 83.30 |
| BERT-BiLSTM-ATT | 81.55 | 84.97 | 86.78 | 83.33 | 83.33 | 83.33 | 83.33 | 83.33 |
| BERT-BiGRU-ATT | 17.62 | 85.17 | 86.58 | 85.88 | 83.33 | 83.33 | 83.33 | 83.33 |
| BERT-CNN-LSTM | 34.85 | 85.90 | 86.32 | 87.64 | 86.55 | 87.14 | 85.43 | 87.21 |
| DC-CBLA | 25.90 | 86.71 | 86.78 | 84.94 | 89.35 | 89.53 | 88.09 | 89.46 |

## Comparative experiments of word vector models

Travel review utterances contain plenty of metaphors, sarcasm, and overly colloquial expressions, and different text vectorization methods have a direct impact on the overall model statistic. In order to investigate the impact of various word vector models on the classification performance of tourism text, we also conducted comparison experiments on the tourist attraction and hotel datasets.

1. W2V-CBLA: The Word2Vec static word vector model is used as the embedding layer and then input into the dual-channel model for training.

**Table 7  Variation in accuracy values for tests at different epochs on the travel attraction dataset.**

| Model | Epoch | | | | | | | |
|---|---|---|---|---|---|---|---|---|
| | 1 (%) | 2 (%) | 3 (%) | 4 (%) | 5 (%) | 6 (%) | 7 (%) | 8 (%) |
| BERT | 24.59 | 91.47 | 92.11 | 80.10 | 80.10 | 80.10 | 80.10 | 80.10 |
| BERT-CNN | 20.30 | 92.62 | 94.42 | 94.44 | 94.95 | 94.49 | 94.20 | 90.01 |
| BERT-LSTM | 28.22 | 92.41 | 80.40 | 80.40 | 80.40 | 80.40 | 80.40 | 80.40 |
| BERT-BiLSTM | 63.16 | 94.25 | 80.50 | 80.50 | 80.50 | 80.50 | 80.50 | 80.50 |
| BERT-BiLSTM-ATT | 79.02 | 79.42 | 94.05 | 93.90 | 94.56 | 94.74 | 94.66 | 94.30 |
| BERT-BiGRU-ATT | 20.81 | 92.07 | 89.86 | 92.65 | 93.71 | 94.44 | 94.38 | 94.24 |
| BERT-CNN-LSTM | 34.51 | 92.34 | 94.55 | 94.91 | 93.85 | 94.53 | 94.22 | 94.76 |
| DC-CBLA | 26.75 | 92.35 | 94.57 | 94.95 | 95.23 | 94.39 | 94.95 | 94.84 |

2. DC-CBLA: This article's model.

According to Tables 8 and 9, our BERT-based word vectorization significantly outperforms the conventional Word2Vec method in terms of performance metrics. Word2Vec provides a static word vector representation where the vector remains the same for any occurrence of a word. Thus, the static word vector scheme is flawed when dealing with words that have multiple meanings. In order to better capture the contextual information of the text, we chose BERT as the word embedding model to calculate the contextual representation of each word. Based on its various semantics, BERT provides different vectorized representations for the same word.

## Hyperparameter tuning

The size of hyperparameters in deep learning plays a crucial role in the model's training process. To enhance the model's performance, we conducted in-depth research on the convolutional window size, learning rate, and dropout settings using the tourist attraction dataset as an example.

Dropout is one of the essential parameters for model training. A suitable dropout rate facilitates model convergence, inhibits overfitting, and enhances model performance. In this research, the dropout rate is set to [0.1, 0.2, 0.3, 0.4, 0.5, 0.6]. Multiple sets of experiments are conducted to determine the optimal dropout rate. The results indicate that the model achieves the most significant effect when the dropout rate is set to 0.2, as shown in Table S1. To improve the neural network's ability to fit the data and ensure the model's generalization performance, a dropout rate of 0.2 is chosen for this experiment.

The setting of the learning rate has a significant impact on the training outcomes and duration. A relatively high learning rate will increase the amplitude of the loss function, making it more challenging to find the optimal global solution and delaying the model's convergence. With a low learning rate, the efficacy of the loss function's learning may be diminished, leading to overfitting. In order to improve the efficiency of this model, various values of learning rate were selected for this experiment. From Table S2, we can see that when the learning rate is 0.00005, the model has the best effect. Therefore, the learning rate is set to 0.00005.

**Table 8   Results of experiments with different word vectorized on tourism attraction data.**

| Model | Accuracy (%) | Precision (%) | F1-score (%) |
|---|---|---|---|
| W2V-CBLA | 93.89 | 95.42 | 96.58 |
| DC-CBLA | 95.23 | 96.53 | 97.05 |

**Table 9   Results of experiments with different word vectorized on tourism hotel data.**

| Model | Accuracy (%) | Precision (%) | F1-score (%) |
|---|---|---|---|
| W2V-CBLA | 86.28 | 86.34 | 91.79 |
| DC-CBLA | 89.46 | 90.91 | 93.86 |

The setting of the convolution window determines the size of the feature information in each extracted piece of data. Therefore, the size of the convolutional window has a relatively major impact on CNN. In this experiment, the filter sizes are set to (2,3,4), (3,4,5), (4,5,6), and (5,6,7). The best results of the model can be seen in Table S3 when the convolution window is (3,4,5). Therefore, this experiment uses a convolution window of size (3,4,5) to extract the feature information of the text.

## CONCLUSION

Online travel platforms are an essential way for visitors to access information and express their opinions, and substantial review texts are generated in the process. Techniques for sentiment analysis can be used to extract opinions from the reviewed text. However, existing approaches have drawbacks, such as inaccurate classification and low efficiency. Based on the characteristics of evaluations in the tourism industry and combining the advantages of CNN, LSTM, and the attention mechanism, this research proposes a BERT pretrained dual-channel hybrid neural network model. The model first encodes the review text into a dense low-latitude word vector matrix using BERT word embedding, then passes through a dual-channel network to extract key global features and deep local features. Finally, stitch the vectors together to obtain the final feature vector representation. Our proposed model is compared with seven other groups of models on tourist attraction and tourist hotel datasets for experiments. The proposed model DC-CBLA achieves the highest precision, accuracy, and F1-score, demonstrating its superiority. The model described in this research increases accuracy but at the cost of time. Future research will focus on improving accuracy while reducing training time.

### Funding

This article and research were supported by the Undergraduate Education and Teaching Research and Reform and Undergraduate Teaching Engineering Project of Chengdu University of Information Technology NO. (JYJG2021104, JYJG2021094), the Cooperative Education Project of Enterprise and School in 2020 NO. (202002230010), and the

Cooperative Education Project of Enterprise and School in 2021 NO. (202101014067, 202101291013). This work was also supported by the Open Project of National Intelligent Society Governance Testing Area NO. (ZNZL 2023A04, ZNZL2023B07), the Science and Technology Program for Overseas Students in Sichuan Province NO. (2022-30) and the Meteorological information and Signal Processing Key Laboratory of Sichuan Higher Education Institutes of Chengdu University of Information Technology. The funders had no role in study design, data collection and analysis, decision to publish, or preparation of the manuscript.

### Grant Disclosures

The following grant information was disclosed by the authors:

Undergraduate Education and Teaching Research and Reform and Undergraduate Teaching Engineering Project of Chengdu University of Information Technology No: JYJG2021104, JYJG2021094.

The Cooperative Education Project of Enterprise and School in 2020 No: 202002230010.

The Cooperative Education Project of Enterprise and School in 2021 No: 202101014067, 202101291013.

The Open Project of National Intelligent Society Governance Testing Area No: ZNZL 2023A04, ZNZL2023B07.

Science and Technology Program for Overseas Students in Sichuan Province No: 2022-30.

Meteorological information and Signal Processing Key Laboratory of Sichuan Higher Education Institutes of Chengdu University of Information Technology.

### Competing Interests

The authors declare that there are no competing interests.

### Author Contributions

- Hanyun Li conceived and designed the experiments, performed the experiments, performed the computation work, prepared figures and/or tables, and approved the final draft.
- Wenzao Li conceived and designed the experiments, performed the computation work, authored or reviewed drafts of the article, and approved the final draft.
- Jiacheng Zhao performed the experiments, prepared figures and/or tables, and approved the final draft.
- Peizhen Yu conceived and designed the experiments, authored or reviewed drafts of the article, and approved the final draft.
- Yao Huang conceived and designed the experiments, analyzed the data, authored or reviewed drafts of the article, and approved the final draft.

### Data Deposition

Code and raw data are available Supplemental Files.

## Supplemental Information

Supplemental information for this article can be found online at http://dx.doi.org/10.7717/peerj-cs.1538#supplemental-information.

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
