# Peer review of "A sentiment analysis approach for travel-related Chinese online review content"

_PeerJ Computer Science, doi:10.7717/peerj-cs.1538_

## Round 0.1 · original submission · Major Revisions

Reviewers such as 1 and 3 raise several valid concerns about the article that needs to be addressed. Below are some of the concerns about clarity and validity.
Firstly, the English used throughout the article is poor and requires significant improvement.
The dataset details are not clearly explained. It would be helpful to summarize the dataset statistics in a table to provide a better understanding of the data used in the experiments. Additionally, the title of the subsection "Experimental parameter" could be renamed to "hyper-parameter tuning" or "model selection," as it is not clear what the term means.
Furthermore, the "Experimental evaluation index" needs to be defined more precisely, and the abbreviations used for TP, TN, and FP should be explained. The list of baselines used in the experiments should also be clarified. A table of detailed results would be helpful to compare the performance of the compared methods, as it is difficult to interpret the performance gap from the figure.
Regarding the validity of the findings, the review suggests that the variations between the accuracy values of the different methods are subtle, making it difficult to determine which method is the best. Additionally, it is unclear which parameters in Table 1 are related to the CNN and which are related to the LSTM.
The review also suggests that comparing the proposed method with a model consisting of only BERT and a classification layer would be useful. Furthermore, the article should include an analysis of the ability of the hybrid architecture to capture the semantics of Chinese expressions and improve sentiment analysis performance compared to individual methods such as CNN and BiLSTM.

·

Basic reporting

This article presents a neural architecture for classifying customer sentiment in Chinese comments about hotels and attractions into positive and negative categories. The architecture combines LSTM and CNN models on top of BERT vectors. Overall, the novelty in this work is very limited, the contribution is insignificant, and it has many inaccurate and ambiguous statements. The experimental setting is poorly designed and described, and the results are not properly presented and discussed. There is a clear inconsistency in the terminology used, and most of the abbreviations in this work are not defined. More importantly, there are some accuracies in some presented equations. More details are provided below:

*****Used English: The English used throughout the article is poor and the article requires significant improvement in numerous locations.
***** Notes about the abstract:
- Line 16: Typically, the term "Bert" is expressed in capital letters, denoted as "BERT"
- Line 16; you mentioned “First, the model uses the Bert model as the word embedding layer to train word vectors.”, what do you mean by training word vectors? Do you mean fine-tuning BERT.
- Line 17: Combine the advantage CNN and LSTM is not sufficiently clear.
- Line 20: “classification is completed by the fully connected layer” do you mean “performed’? Completing the classification gives the impression that the classification has already started at an earlier stage (or layer).
- Line 22: “the proposed DC-CBLA model can effectively improve the classification results”, In order to substantiate.
- the assertion of enhancement through your proposed approach, it is necessary to provide numerical evidence and specify your baseline.
- I recommend that you rewrite the abstract in a more rigorous academic writing style.
- The list of key words is very general and not sufficiently representative.
*****Notes about the Introduction:
- I’m not pretty sure about what do you mean by “There are still some shortcomings and improvements in sentiment analysis methods for the tourism sector.”
- The introduction section of your article is structured as a related work section. It is important to begin by clearly defining the problem and explaining why it is significant. Next, provide a top-down classification of the various approaches that have been proposed and support each class with relevant references. Then, specify the gap and how your proposed method aims to fill it. However, in your article, you simply list some of the related work and provide a summary of each, making it difficult to identify the limitations that you intend to address in -your study.
- You claim in the introduction section (lines: 58-63) that since the Chinese expression are diverse, one neural network is not sufficient to capture them, so you need to combine to neural network (LSTM and CNN), how do you support this claim?
- How do you support your claim that “BERT solve the problem of multiple meanings of words to the maximum extent.” (At line 66)
- The summary of contribution is not quite clear and the novelty in this work is very limited.
-
*****Notes about the literature review
- Line 77: do you mean by “sentiment dictionary” the lexicon-based approaches?
- It is preferred to summarize the related literature in a table.

*****Structure, Figures and Tables
- The article is not well-structured and doesn’t conform to PeerJ standards. for example, “Justify” indentation should be used throughout the article, the paragraphs size and the spacing need serios improvement.
- It is not clear in figure 1 how LSTM and CNN receives the Input from BERT

*****Definitions of all terms and theorems
- It would be helpful to readers if you could define the abbreviation the first time it is used in the text. Some examples are:
o DC- CBLA (Line 16)
o CNN and LSTM (line 62)
o BiLSTM (line 51).
o BiLSTM-ATT (line 156)
o TP, TN, FP; (lines 289-301)
- Too many mathematical terms in the equations (3-11) are not properly defined, e.g., the terms used in the LSTM equations.
- I believed that Equation 10 is incorrect, what is the difference between e and exp(e) and why exp(e) is not used in the denominator?
*****Methodology
- It is sufficient to refer to the original BERT paper rather than providing a detailed explanation of BERT. The same note applies to LSTM.
- You used the term “dual channel” in the abstract and the introduction to describe your proposed method but never used it in the methodology section.
- In the abstract and the introduction sections, you mentioned that you use CNN, and in the methodology, you mentioned TextCNN instead. You need to be consistent throughout the article.
- The idea of using a hybrid method of CNN and LSTM to classify short texts in Chinese is not a new idea. Please refer to the following:
1) “Li, Xuewei, and Hongyun Ning. "Chinese text classification based on hybrid model of CNN and LSTM." Proceedings of the 3rd International Conference on Data Science and Information Technology. 2020.”
2) Zhou, Yancong, et al. "Text Sentiment Analysis Based on a New Hybrid Network Model." Computational Intelligence and Neuroscience 2022 (2022).

Experimental design

- The details of the dataset are not sufficiently clear. It is preferred to summarize the dataset statistics in a table.
- It is better to rename the subsection “Experimental parameter”, as “hyper-parameter tuning” or “Model selection”. The title “Experimental parameter” is not quite familiar.
- Do you mean by “Experimental evaluation index” Evaluation metrics?
- The definitions of TP, TN, and FP are not quite clear, and the abbreviations are not defined.
- The list of baselines is not clearly defined.
- It is hard to compare the performance gap between the compared method from the figure. A table of detailed results is preferred.
- There is no need to present the result of hyper-parameter tuning in the experimental results. If you feel that they may add value to this work, you can at least move them to the appendix.

Validity of the findings

- The variations between the accuracy values between different methods in Figure 5 are very subtle, and it is hard to tell from the figure which method is the best.
- Regarding the model parameter mentioned in Table 1, it is unclear which parameters are related to the CNN and which are related to the LSTM.

Additional comments

- I’m not sure how useful it is to use LSTM with attention and CNN on top of words vector generated by BERT. It is useful to compare this method with a model that consists of BERT and Classification layer and compare it to the proposed method.
- The analysis that shows the ability of the proposed hybrid architecture to capture the semantics of the Chinese expressions and improve the sentiment analysis performance, as you suggest in the article, compared to the individual methods (CNN, BiLSTM etc. ) needs to be included.

Reviewer 2 ·

Basic reporting

no comment

Experimental design

It is recommended to add new references as well in the introduction section.

Validity of the findings

It is recommended to add a comparison table with up-to-date references.

Additional comments

no comment

Cite this review as

·

Basic reporting

no comment

Experimental design

no comment

Validity of the findings

no comment

Additional comments

The paper entitled “A sentiment analysis approach for travel-related Chinese online review content” presents the idea of building a sentiment analysis system for Chinese tourists’ reviews. The main goal of the system is to help businesses to be familiar with customer requirements to provide better services. The issue with the proposed sentiment analysis systems, as the authors claim, is that these systems fail to deal with colloquial reviews, sparse feature dimensions, and metaphors. The authors proposed a new approach that combines the benefits of Bert model with CNN and BiLSTM models. In addition to these models, an attention mechanism is used to promote good features and eliminate or lessen the importance of noisy features. The proposed system is promised and there is a good contribution. The methodology and the experimental side are clear, but there are some English and other aspects that the authors need to do for the paper to be accepted. These are the points that need some modification and updating:
1- In line 51 when the authors mentioned the Huang et al. reference, this in-text citation is not clear. It’s not clear how they used the attention mechanism for dates and why they did it for dates. This citation needs to be rewritten with some clarification.
2- In line 96, there is a grammar error “Liang et al. [11] Introducing …”. This should be Liang et al. [11] Introduced.
3- The sentence between lines 97 to line 99 needs to be rewritten. It’s a too-long sentence and not clear what the author is trying to explain. Also, in line 97, there is the pronoun “we”, which is not convenient here.
4- The abbreviation “RN” refers to what?
5- In line 103 there is a grammar error “proposed using …”. Two verbs in the same place. It’s better to remove “using”.
6- Too long sentence between lines 103 to 105. It’s better to separate this long sentence into two or three sentences.
7- The abbreviation (LDA) in line 107 is not clear.
8- The sentence between lines 114 to 117 is too long. It’s better to separate it into two or three sentences.
9- The sentence in line 117 “ Kim et al. [16] proposed to transform preprocessed word vectors into a two-dimensional word vector matrices and use” has two grammar errors: the “proposed to” and the verb “use” that should be in the past “used”. It’s better to write this way “Kim et al. [16] transformed ….”
10- The abbreviation (RNN) is not clear. This should be short for the recurrent neural network, but the author did not explain that.
11- In line 124, there is a grammar error (not correct tense). The verb “divides” should be divided. The same is in lines 129 and 130 for the verbs proves and propose that should be proved and proposed. Also, the verb “propose an LDA” is in line 135.
12- Line 128 add comma “,” before the sentence “LSTM extracts…”
13- Grammar error In line 162 where the authors used two verbs in one sentence “we use……can better…”. It’s better to rewrite this sentence or add a comma “,” after the word “vectorization” and add which to look like “vectorization, which can better ….”
14- Incorrect use of the preposition “in” in line 182.
15- Incorrect use of conjunction in line 183. This should be “and cost…”
16- In line 183, the authors did explain what “Glove” means.
17- The sentence between lines 190 to 191 is not clear at all.
18- Line 204 the word “Concatenate” should not be capitalized.
19- In Section 4 line 270, in the sentence “More than 36, 000 positive and negative reviews were used for training and validation and 6, 200 were used for testing.” It’s not clear where the 36000 and 6200 came from. Are these the attraction reviews? Please explain that in the paper.
20- Line 300 messing conjunction “and the experiment” or add period “.” And start a new sentence.
21- Not clear abbreviation of “GRU” in line 311.

---

## Round 0.2 · Minor Revisions

Thank you for addressing the previous comments. However, reviewer-1 believes there is still a need for some improvements, such as the language/grammar in the article still requiring substantial improvement. Additionally, there are questions and suggestions regarding the clarity and accuracy of certain statements and terms used in the article. This includes additional information such as datasets, experimental results, and training/testing details, and these are necessary to enhance the credibility of the research. Overall, areas still require minor revision and further substantiation to improve the article's quality.

Please see reviewer-1 comments.

·

Basic reporting

Thanks for the detailed responses to my previous comments. This article has been improved significantly compared to the previous version, but it still needs much hard work before being published in PeerJ Computer Science. Below are my comments on the revised version.
1. Used English: The language used still needs significant improvement. The list of issues I gave you in the first version is just a sample. The article still suffers from ambiguities, typos, grammatical mistakes, and misplaced or missing punctuation. Some of the additional issues in the revised version include (but are not limited to) the following.
• “ to select a more inexpensive”
• By analyzing and processing travelers’ comments, the tourism industry prompts scenic
• 41 spots to identify problems …’
• “This method, which::is based: on: a model of an artificial neural network,”
• analyzed the effects of classification results with different parameters and word vectors.
• “After obtaining the word vector matrix, and then in the hybrid neural network
• layers, we use”

2. New comment. In your answer to comment 4 about the benefit of combining CNN and LSTM, my question is which part is responsible for capturing global semantic features and which one captures the local features. Still, the point needs to be clarified.
3. New comment. In your response to comment 5, the expression “input classification” is unusual. I understand that you mean the classification of the review that has been fed as an input to the network, but it is not usually expressed this way.,
4. New comment about your response to Comment 7. Even though the abstract has been improved, it still has issues. For example, “To further enhance the classification accuracy and applicability of the model,” You didn’t mention any models prior to this sentence you can refer to. Besides (“from Transformers (BERT)”), instead of using this, you can simply say, from “BERT, a transformer-based model”, as you have only used BERT. Additionally, the sentence “Moreover, it indicates that our model for sentiment analysis is reasonable and effective.” is subjective.
5. A new comment about your repose to comment 9, you compare (in the updated sentence) the text SA to the traveling-related reviews, and these are not comparable. You may compare these reviews to other types of reviews. Besides, you need to support this assertion. Personally, I believe the reviews are always subjective.
6. New comment (line 52): What do you mean by “sentiment words”? Do you mean the words of the review, the review and the sentiment are not the same? The sentiment is the opinion expressed through writing a review.
7. New comment (line 64), what do you mean by “After model optimization,:: it was”?

8. New comment about your response to comment 11. In your revised text that you added as comment 11 (from the first review round), you mention that “The fundamental models CNN and LSTM (long short-term memory) have poor feature extraction abilities, cannot extract deep-text sentiment features adequately, and cannot identify critical elements that have a more significant influence on sentiment tendency.” How do you support this claim?!!!
9. New comment. Same note about your response to comment 12, how do you support your claim?
10. Table 1 needs to be supported with more information about the related work, such as the used datasets (names, sizes), the domain and some sort of experimental results.
11. A new comment about your response to Comment 26. The table also should display information about the training/dev/ testing splitting.
12. New comment about your response to Comment 29. I don’t think that using the term “quantity” is accurate here.
13. A new comment about your response to Comment 32. I don’t believe that adding the results of individual epochs makes any difference regarding my concern, it may give us insights about the training behavior, but we care more about the final reported results when comparing the method to the baselines.
14. New comment. There are a LOT of unsupported claims in this article, besides the claims mentioned above; please check this: “It is
15. more advantageous in processing long sequences of text and alleviates the gradient disappearance phenomenon.”
16. The activation function is called ReLU, not Rule (line 363)

Experimental design

-

Validity of the findings

-

Additional comments

-

Reviewer 2 ·

Basic reporting

All the basic reporting is presented in a professional way.

Experimental design

The experimental design is presented in a professional way.

Validity of the findings

All the findings are presented professionally.

Additional comments

It is recommended to add one comparison table for comparing with the up-to-date papers.

Cite this review as

·

Basic reporting

Everything looks good. The authors made the required changes.

Experimental design

Everything looks good. The authors made the required changes.

Validity of the findings

Everything looks good. The authors made the required changes.

Additional comments

Everything looks good. The authors made the required changes.

---

## Round 0.3 · accepted · Accept

Both reviewers have confirmed that the authors have addressed all of their comments.

·

Basic reporting

No additional comments. The authors made the required changes.

Experimental design

No additional comments. The authors made the required changes.

Validity of the findings

No additional comments. The authors made the required changes.

Additional comments

No additional comments. The authors made the required changes.

Reviewer 2 ·

Basic reporting

The authors have provided the comments in a professional way.

Experimental design

This section is done in a suitable and acceptable way.

Validity of the findings

The results are novel enough.

Additional comments

No comments.

Cite this review as